# Microglia–Neuron Crosstalk in Obesity: Melodious Interaction or Kiss of Death?

**DOI:** 10.3390/ijms22105243

**Published:** 2021-05-15

**Authors:** Stéphane Léon, Agnès Nadjar, Carmelo Quarta

**Affiliations:** Neurocentre Magendie, Physiopathologie de la Plasticité Neuronale, INSERM U1215, University of Bordeaux, 33000 Bordeaux, France; stephane.leon@inserm.fr (S.L.); agnes.nadjar@inserm.fr (A.N.)

**Keywords:** microglia–neuron communication, hypothalamus, obesity, metabolism

## Abstract

Diet-induced obesity can originate from the dysregulated activity of hypothalamic neuronal circuits, which are critical for the regulation of body weight and food intake. The exact mechanisms underlying such neuronal defects are not yet fully understood, but a maladaptive cross-talk between neurons and surrounding microglial is likely to be a contributing factor. Functional and anatomical connections between microglia and hypothalamic neuronal cells are at the core of how the brain orchestrates changes in the body’s metabolic needs. However, such a melodious interaction may become maladaptive in response to prolonged diet-induced metabolic stress, thereby causing overfeeding, body weight gain, and systemic metabolic perturbations. From this perspective, we critically discuss emerging molecular and cellular underpinnings of microglia–neuron communication in the hypothalamic neuronal circuits implicated in energy balance regulation. We explore whether changes in this intercellular dialogue induced by metabolic stress may serve as a protective neuronal mechanism or contribute to disease establishment and progression. Our analysis provides a framework for future mechanistic studies that will facilitate progress into both the etiology and treatments of metabolic disorders.

## 1. Introduction

Over the last 25 years, since the discovery of the hormone leptin [1], enormous advancements have been made in understanding how the brain directs changes in feeding behaviors and systemic metabolic pathways to maintain the body’s metabolic needs [2]. This advancement has initiated a consensus on a “brain-centric” view, which now considers that obesity does not only result from the dysfunctional activity of peripheral organs, such as the adipose tissue, but can also originate from the brain [3].

Large-scale genomic studies have demonstrated that body weight excess and the associated systemic metabolic defects can derive from spontaneous mutations that involve genes that are expressed in the central nervous system (CNS) [3]. Moreover, multiple lines of evidence have now demonstrated that hypercaloric diet feeding can trigger obesity and multiple metabolic comorbidities via central mechanisms of action [4,5,6,7].

Most of these mechanisms involve changes in the activity of neuronal networks located in the hypothalamus that influence fuel (food) ingestion, dissipation, or storage [8,9,10]. Brain-to-periphery axes controlled by this brain region allow maintenance of constant levels of energy reserves, or the so-called condition of energy balance. The melanocortin circuit, which is formed by different neuronal populations that respectively express the peptidergic precursor pro-opio-melanocortin (POMC) or the neuropeptide agouti-related protein (AgRP), plays a crucial role. When the synaptic plasticity of this circuit is impaired—for instance, in response to prolonged feeding with hypercaloric diets—this neurobiological defect can lead to energy balance dysregulation [5,6], and, therefore, to the establishment or the progression of diet-induced obesity (DIO) [7,11].

The pivotal role played by the hypothalamus in the etiology of obesity and its associated sequelae may not seem surprising given that this brain area has long been known to influence food intake and energy handling. More surprisingly, however, non-neuronal cells may be the key piece of the puzzle, at least based on more recent information.

Astrocytes have classically been known to participate in the so-called “tripartite synapse”, e.g., the integrated functional unit whereby these glial cells communicate with two or more neurons to accommodate their synaptic transmission [12]. Within this cell-to-cell process of communication, however, microglial cells are emerging as key players given their influence on the formation, function, plasticity, and elimination of synapses [13,14].

During overfeeding, microglial cells accumulate in the hypothalamus [15] (“microgliosis”), undergo morphological activation, and augment the production of proinflammatory cytokines and other neuronally secreted factors that potentially interfere with synaptic transmission [16]. These changes seem to occur specifically in the hypothalamus, but not in other brain areas [16], suggesting that a “healthy” microglial–neuronal crosstalk may be crucial for hypothalamus-based regulation of energy balance, whereas a defective process of intercellular communication may promote neuronal dysfunction, energy imbalance, and ultimately obesity.

Our understanding of the role of microglia–neuron interactions in modulating neuronal function in extrahypothalamic brain areas has significantly improved over the last decade [17]. However, the molecular and cellular underpinnings of such intercellular dialogue at the level of specific hypothalamic circuits involved in body weight control have only begun to emerge.

Here, we will discuss recent evidence that sheds new light on the cellular and molecular mechanisms implicated in microglia–neuron communication in hypothalamic circuits that are relevant for body weight and food intake regulation. We will interrogate whether a defective dialogue between hypothalamic microglia and neurons may be causally linked with the etiology of DIO, while also highlighting the remaining outstanding questions on how this intercellular cross-talk operates to maintain energy homeostasis in the whole body.

## 2. The Emerging Neuroimmune Theory of Obesity

Bidirectional interactions between the immune system and the CNS enable immunological, physiological, and behavioral responses. Upon activation by infection or injury, for instance, peripheral cells of the innate immune system synthesize and release cytokines that serve as immune mediators. These peripheral signals travel to the brain and act through complex mechanisms of microglia–neuron communication, ultimately leading to non-specific symptoms of infection, including lethargy or listlessness, amongst others, which are commonly referred to as the “sickness response” [18].

In animal models of DIO, enhanced sickness behavior is observed after a single administration of the bacterial lipopolysaccharides (LPSs) [19]. This phenotype is associated with increased production of proinflammatory mediators in the hypothalamus, including tumor necrosis factor alpha (TNFα), interleukin 1 beta (IL-1β), interleukin 6 (IL-6), the inhibitor of nuclear factor kappa alpha (IκBα), and cyclooxygenase-2 (COX2) [20], thereby suggesting that whole-body neuroimmune axes coordinated by CNS microglia undergo defective regulation in obesity.

In addition to responding to injury or infections, CNS microglia can be also activated by dietary or hormonal factors that influence whole-body energy handling, including, for instance, lipids [21,22], carbohydrates [23], or the hormone leptin [20,24]. This suggests that microglial cell activity may play a critical role in maintaining systemic energy homeostasis. After long-term feeding with hypercaloric diets, an increased number of activated microglial cells has been consistently observed across different studies at the level of the mediobasal hypothalamus, where POMC and AgRP neurons reside [4,25,26,27,28,29,30,31]. These microglial changes are accompanied by the upregulation of cytokine-signaling-related pathways in hypothalamic neurons [4,25,26,27,28,29,30,31], and are not a rodent-specific phenomenon. Indeed, human obesity is also associated with radiological signs of gliosis [4,32,33]. Moreover, histological analyses of post-mortem brain tissues obtained from obese individuals has revealed the presence of microglial cells with aberrant morphologies at the level of hypothalamic areas [34]. Accordingly, body mass index (BMI) positively correlates with microglial soma size in subjects affected by type-2 diabetes (T2D), which is one of the major glucometabolic complications of obesity [35].

Based on these accumulated findings, one emerging theory in the field is that the exacerbated microglial activity induced by overfeeding may sustain and propagate a condition of low-grade chronic inflammation at both the central (hypothalamus) and systemic level, which contributes to the establishment and the progression of DIO and its associated glucometabolic complication. Several observations support this model: (i) inhibiting expansion of CNS microglia expansion in DIO mice hindered diet-induced weight gain and prevented central and peripheral inflammatory responses induced by overfeeding [15]; (ii) pharmacologically depleting microglia or selectively restraining microglial inflammatory signaling pathways in the hypothalamus sharply reduced hypothalamic microgliosis while also limiting diet-induced hyperphagia and weight gain in a murine model of DIO [36]; (iii) several molecules and pathways have been identified as candidate mediators of hypothalamic inflammatory reactions observed during chronic feeding with a hypercaloric diet (for a detailed review on this topic, see [37]). The targeted disruption of these pathways in a hypothalamus-specific manner, by pharmacologic or genetic means, limits the extent of DIO and its main systemic metabolic defects, e.g., impaired systemic glucose homeostasis [26,27,28,29,30,38].

Of note, emerging microglia-directed neuroimmune responses induced by diet and/or obesity share elements of similarity with other “classic” neuroimmune axes. For instance, a feature of the sickness behavior is the enhanced entry of peripheral immune cells into the CNS [18]. Likewise, during high-fat diet (HFD) consumption, changes in the peripheral and central immune systems are intimately interconnected.

In DIO mice, blood-borne monocyte-derived cells can extravasate the vasculature and enter the brain at the level of the hypothalamic arcuate nucleus (ARC) of the hypothalamus [36], which contains a leaky blood–brain barrier [39]. As they accumulate, these infiltrating cells resume a typical microglia morphology and contribute to propagating the diet-induced inflammatory response [36].

Additionally, diet-induced hypothalamic inflammation, and, similarly, other microglial neuroimmune responses, can have both positive and negative downstream neuronal effects. After brain injury or pathogens invasion, an initial microglial activation is typically observed and is necessary for neuroprotection, but increased or prolonged microglial activity can have neurotoxic effects [40]. Likewise, short-term microglial responses to diet-induced metabolic stress may initially be protective for neurons that are in strict communication with these glial cells, whereas in the long term, the communication process may become maladaptive, thus negatively influencing neurotransmission [41]. Supporting this view, ARC microglial activity and the resulting local increase in proinflammatory cytokine levels observed during DIO progression follow a dynamic pattern, as they occur rapidly during the first hours [42] and days [4] of overfeeding, are normalized for a few weeks during the dietary administration, and rise again thereafter [4]. Moreover, the establishment of a “low-grade” chronic microglial activation state in response to prolonged feeding with hypercaloric diets is often associated with hallmarks of hypothalamic neuronal dysfunction in animal models after long-term feeding of hypercaloric diets, including loss of synapses [5], impaired responsiveness to metabolic hormones [7,43,44], altered intracellular organelles function [4,29,45,46], and possibly cell death [47,48], although the latter was not observed in all of the studies [49].

Thus, an intimate and dynamic relationship links exacerbated hypothalamic microglial activity with hypothalamic neuronal dysfunction and DIO progression. Such a neuroimmune basis of obesity may either originate from or lead to a defective process of communication between neurons and surrounding microglial cells.

## 3. Role of Microglia in the Remodeling of Neuronal Networks in Obesity

Microglia can phagocytize neuronal spines, terminals, and cell bodies, and errors in this process can lead to impaired neuronal activity and possibly to cell death [13,50]. In response to pathogen invasion or neuronal damage, phagocytosis of microorganisms, dying cells, or neuronal debris from microglia occurs during the resolution phase of inflammation and involves the release of cytokines and lipid mediators that exert anti-inflammatory and prorepair properties.

Microglia can also remove synaptic elements from neuronal cell bodies, a process that allows the protection of neurons from excitotoxicity, e.g., the neuronal death caused by the overactivation of excitatory amino acid receptors [13]. Early in the course of a disease’s process, it might be useful to strip away dying synapses so that healthy synaptic elements can take over. However, as the insult to the CNS persists, abnormal synapse removal can translate into neurodegenerative cellular states [13].

Consumption of dietary fat has been proposed to induce apoptosis of neurons in the arcuate nucleus, paraventricular nucleus, and lateral hypothalamus in mice [47,48], and such a neuronal loss is often paralleled by microgliosis [47,48], which makes one wonder if diet-induced neuronal death and exacerbated microglial activity in the hypothalamus are the two faces of the same phenomenon. However, the observation that DIO is associated with hypothalamic neuronal death has not been reproduced by all the studies [49]. This inconsistency may be due to differences in the diet-composition, caloric content, length of diet exposure, and technical approach used to assess neuronal counts across these studies [7]. Hence, only addressing whether and how these different contributing factors impact microglial phagocytic capacity and, eventually, hypothalamic neuronal death may provide a final answer.

Irrespective of this dispute, changes in microglial activity induced by overfeeding may have more “subtle” consequences on downstream neurons—for instance, on their synaptic transmission. LPS-mediated activation of microglial cells evokes excitatory responses in hypothalamic POMC neurons and inhibitory responses in AgRP neurons [51]. HFD-induced obesity is associated with the synaptic reorganization of hypothalamic POMC and AgRP neurons [5], which present fewer total synapses on their perikaryal [5]. These synaptic changes are intimately associated with the presence of increased hypothalamic microglial activity [5], which suggests that microglial-mediated effects may be directly implicated, although no direct causal evidence supporting this model is yet available.

Changes in microglial-mediated phagocytosis of synaptic elements (or microglial “pruning”) seem to occur in response to overnutrition in extrahypothalamic areas—for instance, in the hippocampus. In a DIO animal model, low-grade chronic inflammation induced by a hypercaloric diet was associated with reduced cognitive function and with increased microglial pruning [52,53]. Memory impairment observed in these animals occurred along with reductions in hippocampal dendritic spines [53], and both the memory defects and the loss of neuronal spines could be rescued through pharmacological manipulation of microglial phagocytosis [53].

Whether or not microglia-mediated pruning contributes to the specific hypothalamic neuronal dysfunction observed during DIO has not yet been directly tested, but this hypothesis is supported by a series of indirect observations. For instance, in an animal model of obesity and type 2 diabetes (db/db mice), which was characterized by impaired hypothalamic neuronal synaptic plasticity [54,55], hypothalamic microglial cells presented a reduced expression of the phagocytic marker cluster of differentiation 68 (CD68) [56]. In this model, the deficit in hypothalamic neuronal function and the resulting obesity phenotype derived from a defective action of the metabolic hormone leptin.

Likewise, transgenic mice with a specific leptin receptor deficiency in myeloid cells (which include microglia) presented an obesity-like phenotype, a reduced number of hypothalamic ARC POMC-expressing neurons, and signs of impaired microglial phagocytic capacity in the hypothalamus [24]. Thus, microglia-mediated synaptic remodeling of hypothalamic neuronal cells may be under the influence of hormonal signals, such as leptin. This implies that alterations in these microglial–neuroendocrine axes may contribute to DIO pathophysiology. Of note, leptin can potentiate the microglial response to LPS in an in vitro system, and this is linked to morphological changes that render the microglia more reactive [57], further suggesting that this hormone may have an important role in mediating microglial functions.

Notably, in addition to influencing synapse homeostasis through their phagocytic activity, microglial cells can also promote synapse formation [13], although no studies have thus far investigated whether changes in microglial-mediated synaptogenesis occur during diet-induced metabolic stress, which represents an interesting angle for future investigations.

Thus, hypercaloric diets may trigger microglial-mediated synaptic remodeling, which possibly disrupts the activity of AgRP/NPY and POMC neurons, thereby contributing to disease development. However, the defective synaptic plasticity observed in the ARC of DIO animal models likely involves synergy among not only neurons and microglia but also astrocytes [5], which are key components of the tripartite synapse. Thus, studying cell-type-specific mechanisms that link diet-induced inflammation with the impaired synaptic plasticity of ARC neurons is an interesting avenue for future research.

## 4. Impact of Fuel Substrates on Microglia–Neuron Communication in Obesity

Microglia are highly dynamic and continuously move around the brain parenchyma in response to chemotactic signals. Under activated states, these glial cells not only display increased phagocytic activity, but also augment the production of proinflammatory and neuromodulatory factors, which are all processes that require large amounts of energy. To meet these continuous and rapid changes in energy demands, microglia express transporters for glucose, fatty acids, and amino acids, the three main fuel substrates. Moreover, these glial cells are also capable of metabolizing alternative nutrients, such as glutamine [58].

The exact identities of the main extracellular influencing factors leading to reactive microglial states during overfeeding are under intense investigation and have not yet been clarified. However, saturated fat (SF) may be a key piece of the puzzle. In animals fed with a saturated fat (SF)-enriched diet, fatty acids, including palmitic acid, are conveyed to the brain after ingestion. They are then taken up by hypothalamic microglia, and ultimately contribute to inducing the hypothalamic inflammatory reaction [22].

Feeding mice with an SF-rich diet augments the expression of the heat-shock protein 72 (Hsp72), which is a chaperone protein that is implicated in cellular stress, in the mediobasal hypothalamus [22]. Such a molecular stress response is independent of the increased caloric intake associated with the SF-rich diet. Indeed, increased neuronal Hsp72 levels were observed by administering milk fat to control mice fed with a standard chow diet, but not following the administration of isocaloric olive oil [22]. Thus, increased SF intake—and not fat intake per se—accounts for the presence of signs of hypothalamic neuronal stress. Of note, depleting microglia via a genetic approach nearly abolishes the marked induction of neuronal Hsp72 that is otherwise seen in ARC neurons following SF-rich administration [22]. This suggests that changes in microglial SF sensing causally contribute to the neuronal dysfunction observed in DIO.

Saturated fat is likely not the only fuel substrate that mediates diet-induced microglial activation and the resulting negative downstream neuronal effects. Indeed, long-term feeding with a high-carbohydrate/high-fat diet (HCHF) also leads to increased hypothalamic microglial activity in mice, which is associated with the elevated production of advanced glycation end-products (AGEs) from hypothalamic POMC and AgRP neurons. AGEs are products of glycoxidation and lipoxidation reactions that originate from overexposure to sugars. These bioproducts are implicated in many degenerative conditions, including diabetes and Alzheimer’s disease [59]. Certain types of AGEs—for instance, the product N(ε)-(carboxymethyl)-lysine (CML)—are mainly produced by neurons during diet-induced cellular stress, whereas their receptors are specifically localized in non-neuronal cells, including, but not limited to microglia [23]. Genetic deletion of CML receptors in mice fed with an HCHF diet resulted in less microglial reactivity in the hypothalamic ARC and also led to favorable antiobesity effects [23]. Thus, a putative model can be proposed whereby HCHF diets—and the consequent combined overload of both lipids and glucose in the mediobasal hypothalamus—trigger excessive production of neuronal metabolites (such as AGEs) that activate surrounding microglial cells. This results in a vicious feedback loop that impairs neuronal functions.

However, what are the main intracellular underpinnings of such a maladaptive cross-talk? A final answer is not yet available, but common intracellular nodes may integrate changes in fuel substrate handling between neurons and microglia. In both microglial and neuronal cells, maladaptive changes in mitochondrial function and cellular fuel utilization were observed after hypercaloric diet feeding, and in both cell types, these changes were, in part, mediated by the mitochondrial protein uncoupling protein 2 (UCP2) [60]. Of note, increased UCP2-dependent activity in either microglial or neuronal cells played a key role in mediating DIO susceptibility in murine models [61,62,63].

Thus, sensing of specific dietary components, such as fat and/or sugar, may lead to exacerbated cellular fuel metabolism in both microglial and neuronal cells located in the hypothalamus via common (UCP2-mediated) molecular pathways that are essential for proper mitochondrial activity. This implies that changes in fuel substrate uptake and/or metabolism induced by diet-induced caloric overload may directly impact the microglia–neuron crosstalk via mitochondria-mediated mechanisms and the subsequent production of yet-to-be identified energy metabolites that contribute to the intercellular communication.

Supporting this view, specific microglial genetic deletion of lipoprotein lipase (LPL), one of the key enzymes involved in cellular fuel uptake, caused mitochondrial dysmorphologies in both microglia and nearby ARC neurons in animals that were fed with a hypercaloric diet [64]. These transgenic mice that specifically lacked LPL activity in microglia also presented higher body weight gain relative to wild-type animals when fed with an obesogenic diet [64].

## 5. Role of Cytokines in Microglia–Neuron Communication in Obesity

Activated hypothalamic microglia produce a variety of proinflammatory cytokines that potentially influence metabolic or behavioral outputs by inducing changes in neuronal activity [65,66,67]. In vitro incubation of hypothalamic POMC neurons with the microglial-produced cytokine TNF-α led to an increased neuronal firing rate, increased intracellular ATP production, and mitochondrial fusion processes [68]. HCHF diet consumption is associated with elevated hypothalamic TNF-α levels [69] and with signs of POMC neuronal mitochondrial stress [68]. Hence, persistently elevated TNF-α signaling during DIO may cause functional impairment of hypothalamic POMC neurons as a result of alterations in neuronal mitochondrial activity [68]. This hypothesis is supported by multiple studies that showed that DIO leads to mitochondrial dysfunction in POMC neurons and that this specific pathogenic mechanism may promote dysregulated neuronal activity and obesity [7]. The molecular underpinnings of TNF-α-mediated neuronal dysfunction in obesity have yet to be fully elucidated, but cytokine-mediated changes in the c-Jun N-terminal kinase (JNK)/activator protein 1(AP-1) and NF-κB signal transduction pathways may have a central role. A detailed analysis of the neuronal molecular pathways implicated in hypothalamic inflammation is beyond the main scope of the present work, but more detailed information can be found elsewhere—for instance, in [37].

Although it is tempting to speculate that proinflammatory cytokines produced by hypothalamic microglial cells in response to overfeeding may globally contribute to the induction of neuronal dysfunction and obesity, this hypothesis appears to be too simplistic given that certain microglia-derived cytokines can have positive effects on metabolism and body weight regulation. For instance, activated hypothalamic microglia can produce the proinflammatory cytokines IL6 and IL-1β [70], which can have favorable effects on energy balance regulation. Administration or overexpression of these cytokines in the brain results in reduced food intake, increased energy expenditure, and decreased body weight gain [71,72,73,74,75]. This is likely due to changes in the activity of neurocircuitries located in the hypothalamus [76,77]. Accordingly, the central application of IL6 led to potent feeding suppression and to improvements in glucose tolerance in DIO mice via the modulation of neurons located in the paraventricular nucleus of the hypothalamus [78]. Thus, different proinflammatory cytokines can have either positive or negative effects on body weight control, which makes one wonder whether diet-induced hypothalamic inflammation is necessarily a pathophysiological mechanism that contributes to disease progression.

The fact that microglia-derived proinflammatory cytokines may induce dynamic downstream neuronal effects is probably part of the answer. For instance, among families of cytokines that orchestrate hypothalamic inflammatory reactions induced by overfeeding, chemokines act by attracting immune cells to the inflamed site [79]. When proinflammatory cytokines activity is enhanced in the brain of an animal, this generally leads to a condition of initial high-grade inflammation and to a transitory reduction in food intake, which is partly due to changes in the activity of hypothalamic neurons [80,81]. Nevertheless, such acute and robust inflammatory responses and the linked food-intake-reducing effect generally tends to diminish or, in some cases, to fade away a few hours after administration [80,81]. The metabolic or behavioral effects observed after short-term hypothalamic cytokines action during hypercaloric diets feeding may reflect a defense mechanism that limits the introduction of unhealthy diets. Nevertheless, this mechanism of “neuroprotection” may become maladaptive in the long term and have the opposite effects, thereby contributing to disease establishment and progression. Accordingly, certain chemokines, such as the C-X3-C motif chemokine ligand 1 (CX3CL1, or ‘fractalkine’), promote obesity when chronically administered in the brain [82], whereas inhibition of CX3CL1 actions in the hypothalamus of DIO mice through an approach of small interfering RNA reduced local inflammation and led to weight loss and to improvements in systemic glucose metabolism [82]. Moreover, the expression of the C-X-C motif chemokine 12 (CXCL12) and its receptors was induced in the hypothalamus of mice fed with an HFD [83]. Such increased CXCL12 activity has been suggested to favor overfeeding by enhancing the synaptic activity of specific neurons located in this hypothalamic area that produce the hunger-promoting neuropeptide enkephalin [83].

From a broader perspective, cytokine-mediated changes in microglia–neuronal communication likely also involve other non-neuronal cells. As previously discussed, DIO leads to hypothalamic astrocyte activation, and following activation, these non-neuronal cells produce a variety of immune mediators, including TNFα, IL-1β, IL-6, or the monocyte chemoattractant protein-1 (MCP-1) [39,84], amongst others.

When cultured in vitro, astrocytes accumulate lipid droplets under a free fatty acid (FFA)-rich environment, while also producing a large amount of proinflammatory cytokines [84]. If cultured microglial cells are treated with a conditioned medium obtained from lipid-laden astrocytes, this leads to a markedly enhanced chemotactic activity of the microglial cells, which is due to MCP-1-mediated actions at the level of the microglial C-C chemokine receptor type 2 (CCR2) [84]. Thus, astrocytes-derived inflammatory signals may have an impact on neighboring CNS microglia, thereby alimenting the hypothalamic inflammatory response. Given that the chemokine MCP-1 disrupts the integrity of the blood–brain barrier (BBB) [85] and that the increased brain infiltration of bone-marrow-derived monocytes/macrophages contributes to the induction of hypothalamic inflammation [36] (see above), this vicious cycle may be also influenced by peripheral immune cells.

In conclusion, a multicellular model can be proposed whereby, in response to hypercaloric diets, astrocytes, local microglial cells, and peripheral monocytes/macrophages co-direct changes in the local production of cytokines and chemokines, which ultimately impact hypothalamic neuronal activity and energy homeostasis in a highly dynamic manner during the progression of DIO.

## 6. “ON/OFF” Communication between Microglia and Neurons in Obesity

While long-term changes in the activity of CNS microglia can impair neuronal functions, the opposite relationship can be also relevant, as discussed herein. However, can hypothalamic neurons be initiators of vicious interactions with surrounding non-neuronal cells in response to diet-induced metabolic stress? If one considers that neuronal cells in mammals can release ON signals that promote microglial recruitment and activity [82,86], this hypothesis seems plausible.

For instance, the energy substrates adenosine 5′- triphosphate (ATP) and adenosine 5′-diphosphate (ADP) can be released from the neuronal cell body via membrane channels or activity-dependent vesicle exocytosis [87,88]. Microglia contain specific purinergic receptors (such as P2Y12) that are expressed at the level of structural interaction sites located between neuronal cell bodies and microglial processes [88]. P2Y12-containing purinergic junctions allow microglia to constantly monitor the neuronal status. During traumatic neuronal events (e.g., brain stroke) these junctions initiate microglia-driven neuroprotective actions in a targeted manner on the neuronal cell body [88]. Moreover, purinergic receptors are implicated in controlling microglial motility and attraction to synapses [89].

Similar mechanisms of spatially controlled neuronal surveillance and protection may take place in hypothalamic neurons in response to diet-induced neuronal stress. DIO leads to POMC neuronal mitochondrial dysfunctions [7,68] and to a higher number of activated microglial cells that are in close contact with POMC neurons [68], e.g., potential sites of P2Y12-mediated communication. However, no studies have, thus far, directly addressed whether hypothalamic neuronal cells display a plastic ability to release purinergic signals that influence microglia recruitment and functions, which represent an interesting area for future investigation.

Of note, microglial cells in the healthy CNS are immunologically more quiescent than peripheral macrophages [90], which raises the question of whether brain neurons may release OFF signals that keep these glial cells in a non-active state. In hippocampal brain slices, the induction of the major histocompatibility complex class II (MHC-II) in microglia, which is indicative of increased microglial immune response [91], is restricted by neurotrophins released from electrically active neurons, for instance, by the brain-derived neurotrophic factor (BDNF [92]).

BDNF-producing hypothalamic neurons play a major role in energy balance regulation in response to the hormone leptin [93]. Accordingly, mutations in the BDNF gene lead to insatiable appetite and severe obesity [94]. Hence, BDNF might be a neuronal OFF signal that maintains a homeostatic process of neuron–microglia communication in the hypothalamus, while defective BDNF production may ultimately lead to exacerbated microglial activity and obesity.

In addition to neurons, microglia can also release BDNF, for instance, in response to the activation of ATP-sensitive purinergic receptors [95]. Moreover, BDNF-producing hematopoietic cells can migrate from circulation to the hypothalamus; they can make direct contact with neurons in response to the feeding status, thereby influencing whole-body metabolic and behavioral responses [96]. Mice with congenital BDNF deficiency specifically in these hematopoietic cells developed hyperphagia, obesity, and insulin resistance [96], suggesting that defective BDNF production from either neuronal and non-neuronal cells located in the hypothalamus may initiate a vicious cycle leading to defective neuronal functions and obesity.

Neurotransmitters may represent an additional layer of reciprocal ON/OFF regulation between microglia and neurons. Microglia can express receptors for the neurotransmitters glutamate, gamma-aminobutyric acid (GABA), noradrenaline, or dopamine (amongst others) in both in vitro and ex vivo systems [97], although evidence for microglial expression of these markers in vivo is still lacking. In vitro treatment of microglial cells with most of these neurotransmitters inhibits LPS-induced release of various proinflammatory factors from these glial cells [86]. Nevertheless, these neurotransmitter-mediated effects on microglia activity are unlikely to be only directly mediated. Instead, these influences could be indirectly influenced by extracellular ATP, which is released in response to changes in neurotransmission [98].

While several neurotransmitters may act as an OFF-signals that keep microglia in a quiescent state, one exception may be the excitatory neurotransmitter glutamate. Glutamate can trigger activation and release of the proinflammatory cytokine TNF-α from microglial cells [99,100]. Faced with proinflammatory stimuli, such as TNF-α, microglia can, in turn, release glutamate [101], and thus probably enter a vicious cycle that leads to the increased generation of both glutamate and TNF-α. The reciprocal control of glutamate production by microglial and neuronal cells may be part of a feedback loop that protects neurons from excitotoxicity. Alternatively, neurotransmitter-mediated cross-talks between neurons and microglia may serve as a defense mechanism that limits exacerbated cytokine action or excessive microglial phagocytic activity [86]. Whether or not DIO leads to defective neurotransmitters-mediated microglia–neurons communication has yet to be directly addressed. Given that animal models of DIO display impaired hypothalamic expression of glutamate transporters [102], understanding how and whether changes in the balance between both ON and OFF signals orchestrate the (dys) functional dialogue between microglia and neurons in obesity warrants further investigation.

## 7. Conclusions and Perspectives

A better understanding of the neurobiological basis of obesity is urgently needed given that this disease may have an origin in the brain [3] and that currently available antiobesity strategies have not yet achieved a transformative therapeutic impact [103].

New pharmacological tools that selectively and safely target hypothalamic neuronal circuits affected by obesity are on the horizon [38,104]. However, the cellular and molecular mechanisms leading to hypothalamic neurons’ dysfunction in obesity have only begun to emerge, and additional research will be needed before functional optimization and clinical translation of these brain-directed agents.

These future efforts toward understanding the neurobiological basis of obesity should consider that hypothalamic neuronal cells are not isolated functional entities, and maladaptive reciprocal interactions between neurons and surrounding microglia cells likely contribute to the establishment and the progression of the disease (Figure 1).

Several outstanding questions related to the extracellular and intracellular mode(s) of action of this process of cell–cell communication remain. For instance, what are the main extrinsic factors that transform a melodious microglia–neuron interaction into a maladaptive process? The caloric content of the diet, diet composition, and duration of diet exposure, may influence hypothalamic neuroinflammatory responses through multiple mechanisms, including changes in gut microbiota composition [32]. Moreover, the overload of certain dietary components, such as fats or carbohydrates, may engage specific cellular metabolic pathways that interfere with mitochondrial function and with the extracellular release of chemical messengers (such as ATP or ADP) influencing the inter-cellular communication.

Of note, both neurons and microglia in the brain may secrete extracellular vesicles (EVs), which are known to contain a broad range of signals possibly involved in the maladaptive microglia–neuron communication, including chemokines, ATP, and inflammation-associated small, noncoding RNA (miRNAs) [105]. Whether or not mitochondrial or EV-mediated mechanisms of communication may contribute to diet-induced hypothalamic neuronal alterations, however, remains to be directly explored.

In addition to being influenced by extrinsic factors, the maladaptive cross-talk between microglia and neurons observed in obesity may then originate from intrinsic variables, including genetic predisposition. Indeed, human subjects carrying common polymorphisms in the Jun N-terminal kinase (JNK) or the melanocortin 4 receptor (MC4R) gene are more susceptible to development of obesity and hypothalamic neuroinflammation [32]. Furthermore, gender may play a key role given that microglia isolated from adult brains present sex-specific molecular features [106,107,108] and that sex differences in microglial activity determine obesity susceptibility in mice [109,110].

Finally, the maladaptive microglia–neuron crosstalk that is observed in response to overnutrition likely involves synergy with other cell types—for instance, astrocytes [5,111], which are key components of the tripartite synapse. Moreover, peripheral monocytes/macrophages, which can extravasate from circulation to the brain during DIO progression, are known to contribute to neuroinflammation and microglia–neuron communication [36]. Tanycytes, which are specialized ependymocytes that form a blood–cerebrospinal fluid barrier in the hypothalamus, may have an additional influencing role [39] due to their ability to fine-tune the brain transport of nutrients and hormones implicated in metabolic control [112,113]. Additionally, endothelial cells located in the hypothalamus are emerging as key cellular mediators of obesity and related metabolic disorders [114] and may therefore contribute to the modulation of diet-induced inflammatory responses in hypothalamic neurocircuitries [37].

While trying to fill in this knowledge gap, cellular heterogeneity may represent a challenging obstacle, but also a relevant aspect. Indeed, microglia and neurons in the hypothalamus form highly diverse cellular clusters [115,116,117,118], and may therefore operate in response to diverse and cell-specific mechanisms of communication. Accordingly, different subpopulations of microglial cells have been found in the mediobasal hypothalamus of rodents and humans [119], and these subpopulations present a high degree of spatial localization and distinct molecular signatures in response to diet-induced inflammatory conditions [119].

Thus, addressing when, where, and how hypothalamic neurons and surrounding microglial cells communicate in response to diet-induced metabolic stress will be challenging, but this effort may ultimately favor progress into both the etiology and treatment of metabolic disorders.

## Figures and Tables

**Figure 1 ijms-22-05243-f001:**
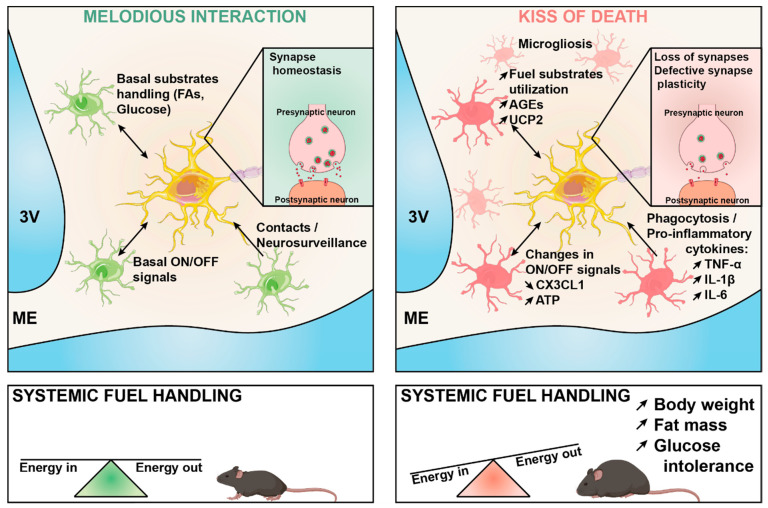
Under physiological conditions (**left**) microglia play a key role in fine-tuning neuronal synaptic function in response to changes in the energy status of the body. This involves different mechanisms including the release of ON-OFF signals, the intercellular production of metabolic messengers in response to the cellular metabolism of fuel substrates, and physical intercellular contacts that are necessary for neurosurveillance. This melodious interaction may become maladaptive during diet-induced obesity (**right**), thereby leading to impaired synaptic plasticity through multiple possible mechanisms, which are highlighted in the figure. The maladaptive cross-talk ultimately results in energy balance dysregulation, obesity, and systemic glucose intolerance. FA: fatty acids; AGEs: advanced glycation end-products; UCP2: uncoupling protein 2; CX3CL1: C-X3-C motif chemokine ligand 1; ATP: adenosine 5′- triphosphate; TNFα: tumor necrosis factor-alpha; IL-1β: interleukin 1 beta, IL6: interleukin 6.

## Data Availability

No new data were created or analyzed in this study. Data sharing is not applicable to this article.

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
