# Peer review of "Microglia–Neuron Crosstalk in Obesity: Melodious Interaction or Kiss of Death?"

_ijms, 2021, doi:10.3390/ijms22105243_

Round 1

Reviewer 1 Report

 “Microglia-neuron crosstalk in obesity: melodious interaction or kiss of death?” by Léon et al. is a timely review on an interesting topic. It covered most of the recent findings made in this research field. I am confident that this review is interesting to general readers. I have only some minor suggestions. 1) There are studies on microglia with obese human post-mortem brain tissues, should these be mentioned and discussed in this review? 2) I would also suggest to improve Figure 1 (or add another figure), the current schema in Figure 1 is quite simplified. I would expect a schema showing under which condition, which factor is involved in melodious interaction or kiss of death, and lead to which consequence at systemic and/or cellular levels. 3) a language polishing might be necessary.

Reviewer 2 Report

1. This manuscript is an interesting review but still needs a complete proofreading in English grammar and some spellings.
2. The author should follow the format rule of the journal. The Figure 1 have no figure legends to explain themselves. The reader even can not identify the cell types in the figure.
3. Some statements are not appropriate scientific language. The author should use a plain scientific language in text. 
4. The subtitles are confused to the readers. The first section is 'Introduction', while the second section is 'Conclusions and perspectives'? 
5. The 'Box 1 : Outstanding questions' is also confusing. It should be transfered to Discussion section. And where are the Box2 and others?
6. Microglia is an heterogenerous cell gourp in CNS. Recent single cell transcriptomics revealed the activation and differentiation of microglia and its influence on microglia-neuron crosstalk. Though the authers are not immunologist, these new finding can not be ignored in a review. This is the major defect of this manuscript.